# Early Circulating Edema Factor in Inhalational Anthrax Infection: Does It Matter?

**DOI:** 10.3390/microorganisms12020308

**Published:** 2024-01-31

**Authors:** Emilie Tessier, Laurence Cheutin, Annabelle Garnier, Clarisse Vigne, Jean-Nicolas Tournier, Clémence Rougeaux

**Affiliations:** 1Département des Maladies Infectieuses, Institut de Recherche Biomédicale des Armées, 91220 Brétigny-sur-Orge, Franceclemence.rougeaux@intradef.gouv.fr (C.R.); 2Institut Pasteur, 75015 Paris, France

**Keywords:** *Bacillus anthracis*, toxins, edema factor, endocytosis, trafficking, enzymatic activity, cAMP

## Abstract

Anthrax toxins are critical virulence factors of *Bacillus anthracis* and *Bacillus cereus* strains that cause anthrax-like disease, composed of a common binding factor, the protective antigen (PA), and two enzymatic proteins, lethal factor (LF) and edema factor (EF). While PA is required for endocytosis and activity of EF and LF, several studies showed that these enzymatic factors disseminate within the body in the absence of PA after intranasal infection. In an effort to understand the impact of EF in the absence of PA, we used a fluorescent EF chimera to facilitate the study of endocytosis in different cell lines. Unexpectedly, EF was found inside cells in the absence of PA and showed a pole-dependent endocytosis. However, looking at enzymatic activity, PA was still required for EF to induce an increase in intracellular cAMP levels. Interestingly, the sequential delivery of EF and then PA rescued the rise in cAMP levels, indicating that PA and EF may functionally associate during intracellular trafficking, as well as it did at the cell surface. Our data shed new light on EF trafficking and the potential location of PA and EF association for optimal cytosolic delivery.

## 1. Introduction

Anthrax is a well-known disease responsible for livestock epidemics and professional diseases and is highlighted in the context of bioterrorism. Three major clinical forms of the disease are described depending on the infection portal: inhalational, cutaneous, and gastrointestinal anthrax, to which an injectional pathology is added. All of them are induced by the spore-forming bacteria *Bacillus anthracis*. At the beginning of the 21st century, atypical strains of *Bacillus cereus* were described in America and Africa, provoking anthrax-like diseases in humans and great apes [1,2,3,4,5,6]. As does *B. anthracis*, these atypical strains produce the two A/B toxins: edema toxin (ET) and lethal toxin (LT). These two major virulence factors are composed of a common binding factor (B), the protective antigen (PA), and two factors with enzymatic activity (A), edema factor (EF) and lethal factor (LF). PA, LF, and EF of the atypical strains share more than 95% protein identity with the factors produced by *B. anthracis* [2]. EF, a calmodulin-calcium-dependent adenylyl cyclase, catalyzes the conversion of ATP to cAMP [7,8]. LF, a zinc-dependent metalloprotease, cleaves and inactivates most mitogen-activated protein kinase kinases [9,10]. The actions of EF and LF induce cellular dysfunction [11,12,13,14,15], leading to paralysis of the immune system at the early stage of infection, whereas, at the late stage, they induce multiorgan failure [16,17].

PA has a key role in virulence. PA (83 kDa) recognizes two major cell transmembrane receptors, tumor endothelial marker 8 (TEM8) [18] and capillary morphogenesis gene 2 (CMG2) [19], the minor receptor and the dominant one, respectively [20,21,22]. A third alternative receptor with low affinity, the β1 integrin, was also described [23]. The native PA (83 kDa) is cleaved by furin-like proteases in the circulation [24,25,26] or at the cell surface after receptor binding [27,28], producing PA (63 kDa) and releasing 20 kDa of the N-terminal portion of the native PA. PA (63 kDa) can then oligomerize in heptamers or octamers [29,30,31], forming a pre-pore to which EF and/or LF bind. Oligomerization leads to relocalization into lipid rafts rich in cholesterol and glycosphingolipids but not in caveolae [32,33,34]. Such partitioning triggers anthrax toxin receptor internalization through clathrin-mediated endocytosis [33]. Thus, the EF and/or LF-loaded pre-pore is endocytosed by the cell. The acidification of endosomes allows a conformational change of PA from the pre-pore to a pore-form [35], which can deliver EF and/or LF to the cytosol by a process called translocation [36,37]. The initial intracellular path of LF and EF after entry is common, up to their delivery to the cytosol [38], where they can exert their respective enzymatic activity. The difference in their destiny occurs after this final step, as EF remains associated with the endosomal membranes while LF diffuses into the cytosol. According to studies, EF can exercise its activity from early [39] or late [40] endosome membranes, explaining the cAMP concentration gradient observed from the center of the cell to the periphery [40].

During the infection process, after the germination of spores, bacilli produce and secrete PA, EF, and LF. In the host, EF and LF can be found as free proteins or complexed to PA oligomers [24,25] but also packaged into exosomes [41] or extracellular vesicles [42]. In the bloodstream, these toxins target white blood cells like neutrophils, lymphocytes, macrophages, and dendritic cells [43,44] and act on epithelial and endothelial barriers [45,46,47], thus favoring bacterial dissemination [48,49]. Experimental studies in various animal models of anthrax infection have shown that circulating LF and EF can be detected in serum and plasma earlier than PA or than the bacteria from which they were produced [50,51,52,53,54,55,56,57,58]. Indeed, inhalational infection of macaques with spores of *B. anthracis* definitely demonstrated the presence of EF and LF in the early phase of infection without any detectable PA or bacteria [50,53]. We also previously demonstrated that the diffusion of LF and EF in plasma after intranasal infection was partially independent of PA [51]. This raises questions about the mechanisms of the passage of these factors across the lung epithelial barrier and the pathophysiological relevance of LF and EF when PA is not or minimally present.

Here, we investigated the cellular trafficking of EF, which has been less studied than LF, in the presence or in the absence of PA. Since the identification of the toxin components in the 1960s, EF has been known to be ineffective in the absence of PA [59], which is necessary for its entry into cells and translocation through the pore for subsequent activity in the cytosol [8,60,61,62]. The entry and trafficking capacity of EF was assessed by producing EF-fluorescent protein chimera containing the monomeric Venus protein. As expected, we observed that EF entered cells in the presence of PA but surprisingly also in the absence of PA in macrophages and epithelial cell lines. Depending on the epithelial cell line, EF endocytosis was pole-dependent. Despite its ability to enter cells alone, EF still needed PA to exert its enzymatic activity. Additional experiments were conducted using a sequential entry of EF followed by PA and showed a retrieval of activity. The association of PA and EF may thereby occur in several sites of the cell and may suggest that EF captured alone could constitute an intracellular reservoir that may matter for *Bacillus anthracis* physiopathology.

## 2. Materials and Methods

### 2.1. Cell Lines and Bacterial Strains

The murine macrophage cell lines RAW264.7 (ECACC 91062702, Salisbury, UK) and J774.1 (kind gift from Pierre Goossens, Institut Pasteur, France) were cultivated in DMEM GlutaMAX supplemented with 10% fetal bovine serum (FBS). The macrophage cell line IC21 (kindly gift from P. Goossens) was maintained in RPMI supplemented with 10% FBS and GlutaMAX. Following cell lines were from ATCC (Manassas, VA, USA). Madin-Darby canine kidney cells (MDCK, CCL_34) and A549 cells (CCL-185) were maintained in DMEM GlutaMAX + 10% FBS + pyruvate. The human pulmonary cell line Calu-3 (HTB55) was maintained in advanced-MEM + 5% FBS + GlutaMAX and Chinese hamster ovary cells (CHOK1, CCL_61) in RPMI + 10% FBS + GlutaMAX.

The competent bacterial strains *Escherichia coli* NEB5alpha (C2987H) and NEBexpressIq (C3037H) from New England Biolabs (Ipswich, MA, USA), were grown in LB medium with the antibiotics ampicillin (100 μg/mL) or kanamycin (50 μg/mL), as required.

### 2.2. Reagents and Antibodies

Endocytosis inhibitors were all purchased from Merck (Darmstadt, Germany) and prepared according to the manufacturers’ instructions: nocodazole (SML1665), bafilomycin A (196000), chloroquine (C6628), and cytochalasin D (250255).

The following antibodies were used: monoclonal anti-His (631212, Takara Bio, Shiga, Japan), goat anti-EF (773L, List Labs, Campbell, CA, USA), goat anti-EEA1 (sc6415, Santa Cruz, Dallas, TX, USA); rabbit anti-CMG2 (ab129004) and rabbit anti-TEM8 (ab19387) were from Abcam (Campbridge, UK); rabbit anti-Rab7 (PA5-78238), donkey anti-rabbit AF568 (A10042), chicken anti-goat AF594 (A21468), and chicken anti-rabbit AF647 (A21443) were from ThermoFisher Scientific (Waltham, MA, USA). All slides and coverslips were mounted using ProlongGlass with NucBlue (P36981, ThermoFisher Scientific).

Q5 DNA polymerase (M0491) and HiFi cloning MasterMix (E2621) were purchased from New England Biolabs (Ipswich, MA, USA) for cloning purposes. Plasmid pQE60 was purchased from Qiagen (Chatsworth, CA, USA) and pET28-mVenus was from our laboratory collection.

Recombinant EF and PA were purchased from Clinisciences (Nanterre, France). Cyclic adenosine 3′, 5′ cyclic monophosphate (cAMP) antiserum, cAMP acetylcholinesterase enzymatic tracer, Ellman’s reagent, cAMP standard, and acetic anhydride used in the enzyme immunoassay (EIA) were all purchased from BertinTechnologies (Montigny-Le-Bretonneux, France).

### 2.3. Cloning, Expression, and Purification of Fluorescent Edema Factor

Cloning. The cya gene from the strain G9241 was used for the construction of fluorescent EF. PCR amplification of plasmid pQE60 and the cya and mVenus genes was performed using Q5 high-fidelity DNA polymerase with designed oligonucleotides (Table 1). The PCR products consisted of an open version of pQE60 with an N-terminal poly-his and a cya-mVenus gene for pQE60-His-cya-mVenus cloning. Cloning was performed following the instructions of the HiFi strategy from NEB, with a 1 h incubation in a thermocycler at 50 °C. After the transformation of NEB5alpha and selection on LB agar plus ampicillin, the sequences were confirmed by sequencing.

Purification. EFvenus was expressed in *E. coli* NEB express Iq grown in LB broth + ampicillin at 37 °C with shaking at 220 rpm until reaching an optical density at 600 nm of 0.6. Following an overnight induction with 0.5 mM of isopropyl-1-thio-a-D-galactopyranoside at 20 °C, with shaking at 200 rpm, the bacteria were pelleted by centrifugation, and the pellets were stored at −80 °C before purification. Pellets were then thawed on ice and suspended in lysis buffer (20 mM Tris, 100 mM NaCl, 25 μg/mL DNAse, 2.5 mM MgCl_2_, 0.1 mg/mL lysozyme, 10 mM imidazole, complete EDTA-free, pH 7.7). Bacterial cells were disrupted by sonication and incubated on ice for 90 min. The supernatants were loaded onto a HisTrapHP column (Cytiva, Marlborough, MA, USA) equilibrated with buffer A (20 mM Tris, 500 mM NaCl, 10 mM imidazole, pH 7.7). The column was washed with buffer A, and the protein was eluted with a 10–400 mM imidazole gradient in buffer A. Fractions containing fluorescent EF were selected after analysis by SDS-PAGE, western blotting (anti-His, anti-EF), and Coomassie staining. They were then concentrated using a spin concentrator 30 MWCO centrifugation unit (Merck). A supplemental purification was carried out by gel filtration using a Superdex S200 increase 10/300 GL column (Cytiva) with buffer B (20 mM Tris, 50 mM NaCl, pH 7.7). Fractions were selected as described above and concentrated using a spin concentrator 50 MWCO centrifugation unit. The protein concentration was calculated by measuring the absorbance at 280 nm. A schematic representation of the fusion protein EFvenus is provided in Appendix A.

### 2.4. Effect of Inhibitors on Cell Viability and Endocytosis Pathways

A549 cells and RAW254.7 macrophages were treated with various concentrations of the compounds for 90 min at 37 °C. Cytotoxicity was tested by crystal violet assay (Appendix A). Briefly, cells were incubated with a 0.2% crystal violet solution for 15 min and washed with water until all excess colorant was removed. Any remaining water in the wells was eliminated, and 200 μL of 0.5% Triton X-100 was added to release the stain from the cells. The absorbance at 595 nm was measured using a Biorad spectrophotometer (Biorad, Hercules, CA, USA). Cytotoxicity was calculated from the absorbance values normalized to the negative control (100% viability) and is expressed as the percentage of cell viability. Each experiment was carried out at least in triplicate.

Inhibitors, their known modes of action, the optimal concentration and the related viability are presented in Table 2.

The effect of the pharmacological inhibitors (cytochalasin D, nocodazole, bafilomycin A1, chloroquine) on the entry and trafficking of EFvenus ± PA in RAW264.7 and A549 cells was determined by pretreating the cells in 96-well plates (0.1 × 10^6^ cells/150 μL) with each compound. After 30 min, cells were exposed to 100 nM EF ± 300 nM PA. After 1 h of challenge, cells were analyzed by confocal microscopy.

### 2.5. Confocal Microscopy

Cells were plated at 0.1 × 10^6^ cells/150 µL on glass slides equipped with a 96-well chamber and allowed to attach overnight. After a 1 h (macrophages) or 90 min (epithelial cells) incubation with 100 nM fluorescent EF ± 300 nM PA in the appropriate medium, wells were washed with PBS, fixed with 4% formaldehyde, and washed again before mounting with ProlongGlass/NucBlue and a coverslip. For transwell experiments, epithelial cells (0.1 × 10^6^ cells/200 μL) were seeded in the upper chamber of a 24-well transwell plate (PET, 0.4 μm). Cells were then incubated with 100 nM fluorescent EF ± 300 nM PA in the appropriate medium for 90 min on the apical (150 μL) or basolateral side (450 μL). Then, cells were washed twice with PBS, fixed in 4% formaldehyde, and the membranes mounted on glass slides. Depending on the experimental conditions (imaging of early or late endosomes), cells were then incubated with fluorescent primary antibodies overnight at 4 °C and secondary antibodies for 1 h at room temperature. Images were acquired by confocal microscopy, ZEISS LSM800 (Carl Zeiss, Oberkochen, Germany), at 40× for A549 and 63× for RAW264.7, and processed using ImageJ software version 1.48C.

### 2.6. Flow Cytometry Analysis

RAW264.7 and A549 cells were seeded in 96 well plates and allowed to attach overnight. Cells were incubated with EFvenus (100 nM) ± PA (300 nM) at 37 °C for the indicated times. Cells were then collected by trypsinization, washed with PBS, stained with LiveDead blue (30 min, 4 °C), washed again with PBS + 2% FBS, and fixed in 4% formaldehyde (10 min, room temperature). Cells, finally in PBS + 2% FBS, were subjected to flow cytometry analysis on a CytoFlex LX instrument (Beckman Coulter, Brea, CA, USA). Data were analyzed using Kaluza software (version 2.1).

### 2.7. cAMP Assay

Intracellular cAMP concentrations in cell lysates were measured by EIA following the manufacturer’s instructions (Cayman Chemical, Ann Arbor, MI, USA). Briefly, the standard curve ranged from 0.3 pmol/mL to 250 pmol/mL and was prepared using untreated cell lysates. ATP interference was eliminated by chemical reaction, and acetylation of cAMP was performed to increase the sensitivity of detection. Absorbance was recorded at 405 nm using a spectrophotometer microplate reader iMark (Biorad).

### 2.8. Statistical Analysis

Statistical analysis was performed using GraphPad Prism software (version 4.0; GraphPad Software, San Diego, CA, USA). Depending on the experiment, statistical significance was determined by the Mann-Whitney test or two-way ANOVA, Bonferroni multiple comparison test. The threshold for significance was set to *p* < 0.05.

## 3. Results

### 3.1. EFvenus Fluorescent Chimera Alone Could Be Found Inside Cells

#### 3.1.1. The Chimera EFvenus Shows Signal Specificity

We followed the entry of EF in the presence or absence of PA by producing a fluorescent EFvenus with a 6xHis tag at the N-terminus. The EF protein corresponded to the mature form naturally secreted by *B. anthracis*, fused to the fluorescent mVenus protein (Appendix A). The His tag allows easier purification and was previously shown to be associated with an increased affinity of EF for the PA63 pre-pore [64,65]. Fluorescent protein mVenus was selected owing to its high brightness and improved acid resistance compared to other common fluorescent proteins [66,67], as the intracellular fate of EF is of interest.

In order to control that the entry of EFvenus was due to a specific recognition of the EF part of the chimera, three distinct experimental methods were employed.

For the first method, different cell lines of murine macrophages and the human epithelial cell line A549, previously used for anthrax toxins studies [33,68,69,70], were exposed to 100 nM of EFvenus ± 300 nM of PA for 1 h at 37 °C. The fluorescence of EFvenus was observed in these cells under confocal microscopy in the presence of PA but also in the absence of PA (Figure 1A,B). To control the fluorescence specific to EFvenus, cells were exposed to the fluorescent mVenus protein associated with a His tag in the presence or absence of PA (Figure 1A). Under this condition, in all the cell lines analyzed, no fluorescence signal for mVenus could be observed contrary to the bright signal obtained for EFvenus alone or EFvenus + PA.

For the second method using the same protocol in RAW264.7 macrophages and A549 cells, the signal was quantified by flow cytometry (Figure 1B). In RAW264.7 cells, less than 5% of cells were positive when exposed to mVenus alone, while 100% of cells displayed fluorescence for EFvenus, with or without PA. In A549 cells, less than 10% were positive when exposed to mVenus alone, and 80% to 100% of cells were positive when exposed to EFvenus alone or with PA, respectively.

In the third method, the entry of commercial EF, not fused to any fluorescent protein, was also confirmed by Western blot with and without PA, showing an increase in quantity owing to exposure time (Figure 1C). A similar profile was observed for EFvenus with or without PA.

Thus, the EFvenus chimera displayed a specific fluorescent signal and a behavior not dictated by the fusion to the mVenus protein. It showed capability to enter into cells in the presence of PA.

#### 3.1.2. EFvenus Alone Enters Cells and Shows Pole-Specificity

Confocal analysis showed a clear cytoplasmic signal of EFvenus with PA but also without PA in all macrophage cell lines tested and in A549 epithelial cells after 1 h of exposure at 37 °C (Figure 1A). Using flow cytometry, the fluorescent signal was detected for both conditions in macrophages RAW264.7 and A549 cells (Figure 1B). However, if EFvenus seemed to be able to enter A549 epithelial cells and RAW264.7 macrophages approximately at the same extent as EFvenus + PA did (Figure 1A), quantitative measurements by flow cytometry showed approximately a 5% reduction in the number of positive RAW264.7 cells, and a reduction of 20% of positive A549 cells in the absence of PA (Figure 1B). Despite only 5% of reduction of EFvenus-positive RAW264.7 cells in the absence of PA, the median intensity of fluorescence (MFI) diminished by 20% compared to the MFI in the presence of PA (Figure 1B).

In order to define if the concentration of EFvenus influenced cell entry in the absence of PA, macrophages and epithelial cells were exposed to increasing concentrations of the chimera with or without PA (Figure 2). In RAW264.7 macrophages, a signal for EFvenus was detected from 12.5 nM to 100 nM. Moreover, a gradual increase in concentration of exposure induced a similar gradual increase in the intracellular signal of EFvenus, regardless of the presence of PA. In epithelial cell lines A549 and CHOK1, the signal was observable from 50 nM to 100 nM of EFvenus with or without PA. CHOK1 displayed an elongated phenotype in the presence of EFvenus + PA, as described in the literature [8,61].

Complementary signal detection by flow cytometry was conducted to obtain quantitative results. RAW264.7 or A549 cells were exposed to 1 nM, 10 nM, or 100 nM of EF venus with a fixed concentration of PA or without PA (Figure 3). In each cell line, signal detection rose with the increase in EFvenus concentration. A statistically significant signal augmentation, compared to unexposed control cells, was detected from 1 nM to 100 nM independently of the presence of PA in RAW264.7 macrophages (Figure 3A). Results in A549 epithelial cells differed: in the absence of PA, the signal was significantly different from control cells only at 100 nM of EFvenus, but in the presence of PA, the signal was significative from 1 nM to 100 nM of EFvenus (Figure 3B). In macrophages, less than 4% of cells were EFvenus-positive at 1 nM; this level reached 40% at 10 nM and 100% at 100 nM, independently of PA. In epithelial cells, fewer cells were positive, with less than 2% and 12% of the cells at 1 nM and 10 nM, respectively, in the presence of PA, and 65% of cells at 100 nM with or without PA.

Data from confocal microscopy and flow cytometry were concordant with a gradual signal intensity associated with the concentration of the chimera EFvenus (Figure 2 and Figure 3). These experiments enabled the define an optimal working concentration of EFvenus. Entry was more important in RAW264.7 macrophages than in A549 cells. However, only a few cells were positive for EFvenus signal at 1 nM and 10 nM even with PA when quantified by flow cytometry, and observation under confocal microscopy needed more than 50 nM EFvenus, which was barely sufficient for epithelial cells. Consequently, all following experiments have been performed with 100 nM of EFvenus, allowing quantification by flow cytometry and displaying a detectable signal in confocal microscopy in macrophages and epithelial cells.

#### 3.1.3. EFvenus Displays a Pole-Specificity in Epithelial Cells, Independently of the Presence of PA

When EFvenus entry with and without PA was studied in other epithelial cell lines seeded onto slides, no signal could be detected for some of them (MDCK and CALU-3). A previous study using the T84 epithelial cell line showed that entry of EF in the presence of PA was dependent on the cellular pole [71]. Thus, entry in epithelial cells was investigated using transwell plates, where access to the apical or basolateral pole was allowed in the same experimental conditions as for macrophages. A fluorescence signal was detected here under confocal microscopy.

The addition of EFvenus ± PA to the upper chamber (apical side) resulted in fluorescence for EFvenus and EFvenus + PA in CHOK1 and A549 cells only, whereas no signal was detected for Calu-3 or MDCK cells (Figure 4). Conversely, the addition of EFvenus ± PA to the bottom chamber (basolateral side) resulted in fluorescence only in MDCK and Calu-3 cells (Figure 4) but not in CHOK1 or A549 cells. Intracellular signal for EFvenus was detected for each cell line according to one unique pole, with or without PA.

Thus, PA appeared to be dispensable for the entry of EFvenus in various cell types, and the pole of entry for EF is the same, in the presence or not of PA: the apical side for CHOK1 and A549 cells and the basolateral side for MDCK and Calu-3 cells.

### 3.2. Entry of the Fluorescent EFvenus Chimera Is Mediated by a Dynamic Process

#### 3.2.1. EFvenus Enters Cells Rapidly

Regarding results from confocal microscopy (Figure 1A, Figure 2 and Figure 4), flow cytometry (Figure 1B and Figure 3), and western blot (Figure 1C), EFvenus was found in cells even in the absence of PA.

The kinetics of cell entry by EF in the presence of PA was proved to be a rapid phenomenon [8,38,40,71,72]. In order to establish if the intracellular presence of EFvenus in the absence of PA followed a similar dynamic process, a quantitative time-course of EFvenus exposure was conducted on RAW264.7 macrophages and A549 epithelial cells. The number of fluorescent cells (i.e., displaying EFvenus entry) and their median fluorescence intensity (MFI) (i.e., indicative of the relative quantity of intracellular EFvenus) were quantified during 2 h at 37 °C by flow cytometry. The fluorescence signal was normalized to the signal obtained for EFvenus + PA at the later timepoint, a condition used as a positive control for cell entry because of signal saturation (Figure 5).

In A549 cells, the entry of EFvenus ± PA followed a linear profile from 2 to 40 min, the time at which the maximal number of cells was fluorescent for EFvenus without PA (Figure 5A). In RAW264.7, the uptake was faster since all the cells were fluorescent in 10 min (Figure 5B). However, while RAW264.7 cells were equally fluorescent at 60 and 120 min for EFvenus compared to EFvenus + PA in A549 cells, EFvenus required PA to augment the number of fluorescent cells from 85% to 100%. The MFI revealed a similarity between cell types, where a higher amount of signal was collected for EFvenus without PA at 10 min in RAW264.7 and A549 cells, compared to the signal with PA (Figure 5).

Entry was also monitored under confocal microscopy. In-vitro time lapse microscopy objectived the rapid trapping of EFvenus by RAW264.7 cells, which seemed faster in the presence of PA (Appendix A).

These results indicate that EFvenus entered the cells rapidly despite the fusion of a fluorescent protein. The profile of endocytosis was similar for both cell lines, faster in RAW264.7 than in A549 in the presence or absence of PA with a plateau after 40 min.

#### 3.2.2. EFvenus Is Endocytosed and Follows the Same Endosomal Trafficking as Efvenus + PA

Previous studies showed that EF bounded on PA pre-pore was endocytosed by the cell [33,38]. Endocytosis is an active process that can be seen by incubating cells at 37 °C, which allows normal cell activity. Incubating the cells at 4 °C is known to block endocytosis. Macrophages and epithelial cells were exposed to EFvenus or EFvenus + PA at 4 °C or 37 °C. All cell types displayed a fluorescent signal only at 37 °C when exposed to EFvenus or EFvenus + PA (Figure 6). As no fluorescence was observed when cells were incubated at 4 °C, we ruled out the passive uptake of EFvenus without PA and concluded that EFvenus entry alone required membrane trafficking.

It is well known that actin and endosomal acidification are required for the EF + PA complex to enter into cells and for EF to exert its enzymatic activity [29,32,39,73,74]. In order to confirm that EFvenus responded to the same requirement as EF, we preincubated cells with specific inhibitors (Table 2) before exposure to EFvenus ± PA (Figure 7). The concentration of each inhibitor was chosen to minimize cytotoxicity (>75% viability) while retaining inhibitory activity (Appendix A).

Inhibition of actin polymerization by cytochalasin D abolished signal detection for RAW264.7 macrophages and drastically diminished signal in A549 cells. Bafilomycin A and chloroquine, two inhibitors of endosomal acidification, totally erased the signal for both cell lines. Nocodazole, which disrupts the trafficking of vesicles from early to late endosomes by interfering with microtubule polymerization, appeared to reorganize the localization of EFvenus and EFvenus + PA in A549 epithelial cells, leading to a crown of fluorescent dots accumulating at the inner edge of the cells (Figure 7). In RAW264.7 macrophages, the effect of nocodazole on the distribution of the fluorescence was less pronounced, and the chosen concentration was too harmful (51.2% viability, Appendix A) to draw any conclusions.

According to previous studies, the complex EF + PA follows early and late endosome trafficking [38,39]. After incubation of RAW264.7 macrophages with EFvenus in the absence or in the presence of PA for 1 h at 37 °C and immunofluorescent staining of endosomes, the localization of EFvenus was monitored according to time. Under our conditions, EFvenus co-localized with EEA1, an early endosome marker, and with Rab7, a late endosome marker, at 15-, 30-, and 60-min post-incubation, in the presence and in the absence of PA (Figure 8).

We concluded that the entry of EFvenus alone needed cellular activity, functional actin, and microtubule networks and used endosomes to traffic inside cells, as EF did in the presence of PA.

### 3.3. EF Requirs PA for Intracellular Activity

#### 3.3.1. EFvenus Displays an Enzymatic Activity in the Presence of PA Only

EF is an adenylate cyclase that raises the intracellular level of cAMP [8,75]. Quantification of cAMP level previously showed variation according to the cell type and the time of exposure [76]. To determine whether intracellular EFvenus is enzymatically active, we monitored cAMP production in several cell lines at various times after exposure. The level of cAMP was compared to that induced by commercial EF in the presence or absence of PA. Owing to the variations in quantification of cAMP between independent experiments, data were normalized to the total protein of the sample and then to the basal intracellular level in unexposed control cells (mean of basal cAMP level in control in picomole per milligram of protein: MDCK, 1.17 pmol/mg; CHOK1, 0.78 pmol/mg; A549, 0.34 pmol/mg; RAW264.7, 0.28 pmol/mg).

In MDCK cells, an increase in cAMP levels was observed at 90 min for both EFvenus + PA (ETvenus) and commercial EF + PA (ET) (Figure 9A). At the 3 h timepoint, the cAMP level remained higher than the control, with a tendency to decline.

In A549 epithelial cells, there was a small increase in cAMP levels after 90 min for ET and ETvenus (Figure 9B). At 3 h, while the increase in cAMP levels induced by ET was higher than at 90 min, there was no increase in cAMP with ETvenus.

CHOK1 cells were the most susceptible to intoxication in our study, showing around 350% of increase induced by commercial ET. ETvenus also induced an increase in cAMP levels of approximately 230% (Figure 9C). High cAMP levels induced by ET and ETvenus were measured at 90 min and 3 h.

In RAW264.7 macrophages, cAMP levels increased 1 h after the challenge with ET and ETvenus (Figure 9D). After 2 h and 4 h, the levels remained higher than that of unexposed cells and seemed to be stable.

The cAMP response varied depending on the cell type and the time point post-exposure. In the absence of PA, cAMP produced by EF and EFvenus stayed at a basal level similar to unexposed control cells. ETvenus showed enzymatic activity in the different cell lines, highlighting the requirement of PA. Globally, EFvenus was slightly less active than EF in the presence of PA but succeeded in raising the cAMP level.

#### 3.3.2. Sequential Entry of EF and PA Allows an Increase in cAMP Levels

We observed that EFvenus could traffic up to the late endosome with or without PA (Figure 8). We, therefore, investigated whether EF could be rescued by adding PA one to several hours later so that it could consequently display activity.

With regard to the increase in cAMP level in the different epithelial cell lines, the CHOK1 cell line was used since it was the most sensitive in our experimental conditions (Figure 9). The study was also carried out in the macrophages RAW264.7 cells. The commercial EF protein was used since it gave a higher cAMP response than EFvenus (Figure 9). CHOK1 and RAW264.7 macrophages were exposed for 1 h to EF (100 nM), washed to remove EF possibly bound to the cell surface, and exposed for 1, 2, or 3 h to PA (300 nM) (Figure 10). Under these conditions, the increase in cAMP levels was slight but significant for CHOK1 cells after 1, 2, or 3 h of subsequent PA exposure and for macrophages after 2 h and 3 h.

Overall, these results showed that EF could be endocytosed independently of PA, and its enzymatic activity could subsequently be rescued if PA was captured later.

Complementary experiments were conducted by exposing cells to PA first and then adding EF for 1, 2, or 3 h before quantification of cAMP. The level of cAMP was raised in all cases in CHOK1 and after 1 and 2 h of subsequent exposure to EF in RAW264.7 macrophages (Appendix A). Additional analysis by flow cytometry demonstrated the efficacy of two PBS washes since less than 0.5% of cells displayed PA at the cell surface (Appendix A).

## 4. Discussion

Anthrax toxins play a major role in the disease associated with *B. anthracis* and in anthrax-like diseases linked to certain atypical strains of *B. cereus*. Previous studies have highlighted the importance of the PA/LF/EF complex, which enters cells to exert its action [60]. In animal models of anthrax, the enzymatic parts of the toxins, EF and LF, were found in the bloodstream in the absence of PA or bacteria and before the detection of PA and bacteria [50,51,53]. Moreover, the levels and dynamics of PA, LF, and EF production and their dissemination in infected hosts have raised questions about the association kinetics of this complex [50,51,52,53,54,57] and the functional relevance of EF and/or LF alone in pathophysiology.

In order to explore the impact of the early release of EF found in the blood circulation in the absence of its functional partner PA and of bacteria, we provided a functionally active fluorescent EF chimera.

We ensured that the fluorescent EFvenus chimera did not induce a nonspecific response, was able to translocate, and was enzymatically active. Indeed, EGFP and mCherry chimeras of EF and LF were previously used to demonstrate that ET and LT share the same internalization pathway and follow the same time course in BHK cells [38]. Nevertheless, these chimeras were not bright enough for detection under confocal microscopy and needed additional immunofluorescent anti-GFP staining, questioning the existence of the chimera itself. The mCherry chimeras were also not able to translocate. In our conditions, our EFvenus chimera displayed a signal sufficiently bright to be directly observed, opening an opportunity for live-cell imaging microscopy, and was enzymatically active, testifying of the correct translocation through PA pore. Moreover, twice less concentration was used for experiments than employed in the aforementioned trafficking study.

To locate and evaluate EF entry in cells, EFvenus fluorescence was observed under confocal microscopy and measured by flow cytometry. As expected, fluorescence was observed for EFvenus + PA (ETvenus) in all macrophages and epithelial cell lines. Surprisingly, we also showed fluorescence inside cells when they were exposed to EFvenus alone, according to an active cell process similar to ETvenus. Thereby, the addition of a fluorescent tag did not prevent the rapid endocytosis of EF/ET. Although the rapid entry of EF + PA had been previously described [8,38,72], it had never been documented for EF alone.

We demonstrate that the endocytosis of EFvenus involved active cellular cytoskeletal processes ETvenus. Abrami et al. previously showed that actin was essential for efficient PA heptamerization in HeLa cells and was required for the endocytosis of heptameric PA pre-pore [32]. More particularly, inhibition of microfilament function blocked the entry of ET into CHO cells and inhibited 93% of cAMP production by ET [73]. In our study, in A549 cells and RAW264.7 macrophages, actin was involved in the entry of EFvenus in the presence or absence of PA. In previous studies, the involvement of microtubules appeared to depend on the cell line. Thus, there was an inhibition of the ET-induced rise of cAMP levels in HeLa cells treated with nocodazole [40] but not in RAW264.7 macrophages [39]. In our study, nocodazole induced a different spatial distribution of EFvenus and ETvenus in A549 cells, suggesting the trafficking of EFvenus in the presence or absence of PA between early and late endosomes. Nocodazole cytotoxicity on RAW264.7 macrophages made it impossible to draw any conclusions on the role of microtubules in the trafficking of EF ± PA. However, in macrophages, we observed a significant colocation of EFvenus ± PA with EEA1, a marker of early endosomes, and with Rab7, a marker of late endosomes. These observations were concordant with some previous results [39]. We can thus hypothesize that, in our cell culture conditions, both EFvenus and ETvenus were trafficked through early and late endosomes in RAW264.7 macrophages, as well as in A549 epithelial cells.

The enzymatic activity of EF is dependent on the translocation through the PA pore and, therefore, on the acidification of the compartment in which the PA pre-pore is present [36,37,73,77]. In HeLa and RAW264.7 cells, the use of bafilomycin A1, an inhibitor of the vacuolar ATPase proton pump, completely abolishes or markedly reduces EF activity [39,40]. In CHO cells, ammonium chloride and chloroquine were reported to inhibit the cAMP response to ET [73]. We obtained similar results with the use of bafilomycin A1 and chloroquine in A549 cells and RAW264.7 macrophages. The fluorescence associated with EFvenus and ETvenus was markedly reduced in both cell types. Thus, acidification appeared to be still primordial for the trafficking of EFvenus into the cell, as for ETvenus.

Several studies have reported a range of kinetics for ET entry into cells from a few minutes to almost 1 h [8,38,40,71,72]. In our study, the addition of a fluorescent tag to EF in the EFvenus chimera did not prevent its rapid endocytosis from the cell surface. Intriguingly, the EFvenus chimera could enter A549 cells and RAW264.7 macrophages just as quickly in the absence of PA. The dynamics and entry profile of EFvenus in the absence of PA were similar to those of ETvenus, strongly suggesting that the dynamics of the processes of endocytosis could be independent of PA binding. Differences were mainly observed for A549 cells, with a greater entry in the presence of PA at late time points.

Several hypotheses could explain the presence of EF inside cells in the absence of PA.

One could find an explanation through interaction with an alternative receptor, as described in a recent work on angiogenesis and growth of chicken embryos [78]. In this work, they described the binding of EF on the hepatocyte growth factor receptor c-Met in the absence of PA. This receptor has previously been shown to direct the cytoskeleton and differentially control membrane ruffling and, thereby, cell motility, depending on which endosomal compartment it is located in [79]. Experiments could be conducted to determine if cMet interacts with EF in our conditions and if changes in phenotypes or cell behavior occur. Moreover, it has also been demonstrated that EF can interact with lipidic membranes independently of the pH value [80,81]. This lipidic interaction has not been explored so far, but it could explain the location of EF, bound to the cytosolic side of endosomes after translocation [39], and could represent another path for entry in the absence of PA from the cell surface. Such lipid interaction was also described in epithelial cells for the adenylate cyclase CyaA of *Bordetella pertussis* in the absence of its receptor CD11b/CD18 and suggested to be due to the particular lipid composition of the membrane [82]. Lastly, *B. anthracis* secretes extracellular vesicles, which contain a heterogen mix of anthrax toxins’ factors PA, EF, LF, and anthrolysin [42]. These vesicles can enter macrophages [42]. EF, circulating in the bloodstream, could be packaged into these vesicles, providing another path for cell entry by endocytosis of the vesicle independently of PA. This was not unprecedented since CyaA, from *B. pertussis*, uses extracellular vesicles to be internalized by J774A.1 macrophages and CHO-K1 epithelial cells independently of its cellular receptor, thus using a distinct mechanism from the toxin alone [83].

The mechanisms involved in the endocytosis of EF in the absence of PA remain to be explored. However, we can rely on existing data for ET and bacterial toxins.

One evident hypothesis would be the entry of EFvenus due to fluid-phase endocytosis, also called macropinocytosis. This pathway is non-specific, does not require cell contact, and is characterized by a huge amount of fluid and membrane internalization. Constitutive macropinocytosis is effective in immature macrophages, and a growth factor-dependent macropinocytosis is seen in epithelial and endothelial cells [84]. Results obtained here for both cell types showed the rapid entry of EFvenus and EFvenus + PA followed by a plateau, suggesting an overload of saturable endocytosis pathways. However, macropinocytosis in macrophages is nonsaturable [85] and does not occur in A549 cells [84]. Thus, the endocytosis of EFvenus must rely on micropinocytosis.

Several studies described anthrax toxin entry by analyzing PA endocytosis and enzymatic activity of EF or LF. Different endocytosis pathways were evoked implicating actin, lipid-rafts, clathrin-mediated, or caveolae-mediated endocytosis, each in different degrees depending on cell lines and cell types [32,33,71]. Data collected here opens new hypotheses to explore. Indeed, different micropinocytic pathways involving different mechanisms could be responsible for EFvenus entry from the cell surface in the absence of PA: the well-known clathrin and caveolae endocytic pathways, but also clathrin-caveolin-independent pathways [86,87]. Moreover, among the clathrin-caveolin-independent endocytosis pathways, leading to membrane invagination due to particular lipids and membrane-anchored proteins [85,86], the dynamin-independent flotillin endocytic route and the clathrin- and dynamin-independent endocytic CLIC-GEEC pathway can internalize extracellular fluids. Besides, one toxin can use different pathways. Thus, the shiga toxin uses a lipid-dependent mechanism of endocytosis, independent of clathrin, dynamin, and caveolae machinery [88], and a fast-endophilin-A pathway. The cholera toxin employs the CLIC/GEEC, the fast-endophilin-A, and the flotillin-assisted pathway. More experiments are required to define the pathways implicated in EFvenus internalization in the different cell lines studied.

Moreover, transcytosis has been demonstrated for other toxins, such as Shiga and cholera toxin in T84 cells [89,90,91] and botulinum neurotoxin A through intestinal cell lines [92]. The cholera toxin acts on the basolateral side, but entry from the apical pole followed by transcytosis to the basolateral side permits activity [93]. The principal entry of type AB iota toxin was at the basolateral side, but transcytoses of its B part also allowed entry from the apical side [94]. A similar capacity of EF to transcytoses should be explored and could explain EF’s presence in the bloodstream after infection [50,51,52,53].

During inhalational anthrax, the apical side of the epithelium is exposed to the spores that can potentially germinate at the air-liquid interface [70]. In the case of blood dissemination, access to the basolateral side of the epithelium is probably allowed to bacteria and to early diffusing EF and LF factors in the absence of PA [50,53]. Thus, the entry of EF through the two poles of the cells was explored.

Interestingly, entry into epithelial cell lines was pole-dependent in the presence or absence of PA: EFvenus and ETvenus used the apical side for CHOK1 and A549 cells and the basolateral side for MDCK and Calu-3 cells. This phenomenon was described for the toxin CyaA from *Bordetella pertussis*, with basolateral pole-dependent endocytosis in T84 intestinal cells and tracheobronchial epithelial cells [95]. As demonstrated in T84 cells for EF in the presence of PA [71], this may be due to the specific distribution of the anthrax receptor, which is cell-type dependent. It may also be due to different levels of endocytosis processes [84] and the internalization machinery available [63], depending on the cell line and/or cell pole. For example, macropinocytosis in MDCK cells is active at the apical side but not at the basolateral surface [84]. MDCK are renal epithelial cells, so entry of EF from the basolateral side was not surprising as the infection came from blood. Lung cell lines, Calu-3 and A549, are issued from bronchial and alveolar compartments, respectively. Interestingly, they displayed distinct poles of entry, basolateral and apical, respectively, which can be associated with differential susceptibility to infection of bronchi compared to alveoli. Moreover, ET and LT can contribute to late access to the basolateral pole by creating macroapertures [45,46] or by disrupting epithelial barrier integrity [96,97], respectively. Transcytosis of spore or bacteria [98], or EF itself as evoked earlier, could give access to the basolateral pole. Considering that other virulence factors of *B. anthracis* [99,100] could be partners for epithelial cell infection, like anthrolysin O [101,102,103] or phospholipases C [104,105], the early dissemination of EF and LF could be assisted for internalization into cells, helping for access to the right pole of endocytosis and destroying physical barriers such as mucus.

Our results suggest that the plasma membrane is not the only critical meeting place for EF and PA. The endosomal location of EF in the absence of PA, similar to in the presence of PA, may be a second site binding site between EF and PA pore or pre-pore, critical for the subsequent enzymatic activity of EF. The endosomal system is a dynamic and interconnected system [105] and such endosomal encounter after individual endocytosis of two partners has already been observed in drosophila [86].

Indeed, although EF can enter cells alone, PA is still required for its functional activity, as demonstrated so far [8,61]. ETvenus was slightly less efficient than commercial ET in increasing the level of cAMP. This is not unprecedented since it has been demonstrated that fusion with eGFP or mCherry can reduce, delay, or abolish the enzymatic activity of EF when challenging cAMP production in the presence of PA [38,106]. Translocation capabilities were incriminated depending on the fluorescent protein fused. In this way, searching for new chimeras needs to be improved. In accordance with our data suggesting a novel cellular interaction site between EF and PA, the simultaneous entry of EF and PA into the cells was not necessary to increase cAMP levels, a response that reflects their translocation to the cytosol. CHOK1 cells and RAW264.7 macrophages responded to the sequential entry of EF and then of PA. Initial delivery of PA and later delivery of EF was more effective in inducing an increase of intracellular cAMP levels in CHOK1 cells and RAW264.7 macrophages (Appendix A). Complementary data showed that our experimental conditions clearly reduced the percentage of PA surface fixation (Appendix A) despite a low constitutive rate of PA endocytosis in some cells. This result is not surprising, as it was previously demonstrated that a proportion of PA can remain on the cell surface after washing and that PA heptamers were found in endosomes within one hour [33].

However, the only way for EF to be enzymatically active is to translocate from the endosome to the cytosol, a mechanism allowed by acidification and then passage through the PA pore. Here, we demonstrate that EF can enter cells alone but stay inactive, so it is not located in the cytosol. One way to keep EF in an inactive state would be to prevent a late event of PA encounter. Assuming that post-exposure to PA can rescue enzymatic activity after a fusion between an EF-containing endosome and a PA-containing endosome, developing inhibitors of PA binding and endocytosis seems to be a key point against intoxication. Molecules displaying an ability to sequester EF would also be valuable by avoiding any internalization into cells.

Our results indicate that EF and PA can traffic separately inside cells. This could present a number of advantages for bacteria.

EF follows the N-end rule for degradation by the proteasome after ubiquitinylation [107]. Ubiquitin is a signal toward a degradation pathway generating peptides for presentation by the type II major histocompatibility complex (MHCII) [108]. In immature dendritic cells, ubiquitinylated cargos in endosomes are sent to intraluminal vesicles (ILV) and to degradation; in mature cells, ILV is transformed into exosomes [109]. Since internalization of PA pre-pore led to its rapid degradation after internalization [33,110], one could imagine that early diffusion of PA alone could not represent an advantage for infection. It has been reported that LF, in the presence of PA, can either finish into the lumen of early endosomal ILV, from which it can either remain or be released into the cytosol after a back-fusion process [41], or be translocated by the PA pore from late endosomes [110]. Translocation of LF by PA pore into ILV protects the factor from lysosomal degradation [41]. ILV corresponds to membrane invaginations of endosomes. Thus, if EF interacts with the endosomal membrane [80,81], it could also be protected from degradation into ILV, waiting for PA or possibly waiting for a back-fusion event to be released in the cytosol. As observed for LF, EF protected into ILV could be retained in cells for several days, allowing long-term activity, constituting a reservoir [41], waiting for PA to exert its enzymatic activity, and impairing immune cells. Since our study showed that a huge amount of EF can be internalized by phagocytes, this could play a great role in the initial phase of infection. Moreover, ILV can be released out of the cells in the extracellular milieu as exosomes. Exosomes containing LF were demonstrated to transmit active LF to cells by a dynamin-dependent pathway without the need for a PA receptor [41]. In the absence of PA, EF could be internalized by phagocytes and released during their circulation in the bloodstream, ready to be internalized by neighboring cells and to prepare for PA arrival.

Another interesting possibility is that some epithelial cells possess receptors implicated in pathogen recognition and express the MHCII, allowing them to act as antigen-presenting cells [111]. A study showed that alveolar type II cells and primary bronchial epithelial cells do not express the co-stimulatory molecules CD80 and CD86, which are necessary for lymphocyte activation. This leads to an anergic state of T lymphocytes interacting with these pulmonary cells [112,113,114]. Thereby, these cells could potentially present ubiquitinylated EF on their MCHII and induce tolerance from T lymphocytes, contributing to a better preservation and diffusion of circulating EF.

## 5. Conclusions

Overall, our study suggested that the early circulating EF alone observed after intranasal infection does matter. Indeed, we demonstrated in vitro that EF could enter epithelial and macrophage cells and reach late endosomes in the absence of PA. The fluorescent EFvenus chimera used in this study was enzymatically active in cells, inducing an increase in cAMP level in the presence of PA. In accordance with most studies, PA was still required for EF to reach the cytosol and induce an increase in intracellular cAMP levels. Interestingly, pre-exposure to PA, or the sequential delivery of EF and then PA, rescued the increase in cAMP levels. These results signify that the association of PA and EF may functionally occur in the late endosome as well as on the cell surface. Therefore, the EF capacity to independently enter and traffic within cells could serve as a reservoir, retrieving a functional activity later after PA encounters inside the cells. Our data provide a new picture of EF trafficking and the location of PA and EF association for alternative cytosolic delivery.

## Figures and Tables

**Figure 1 microorganisms-12-00308-f001:**
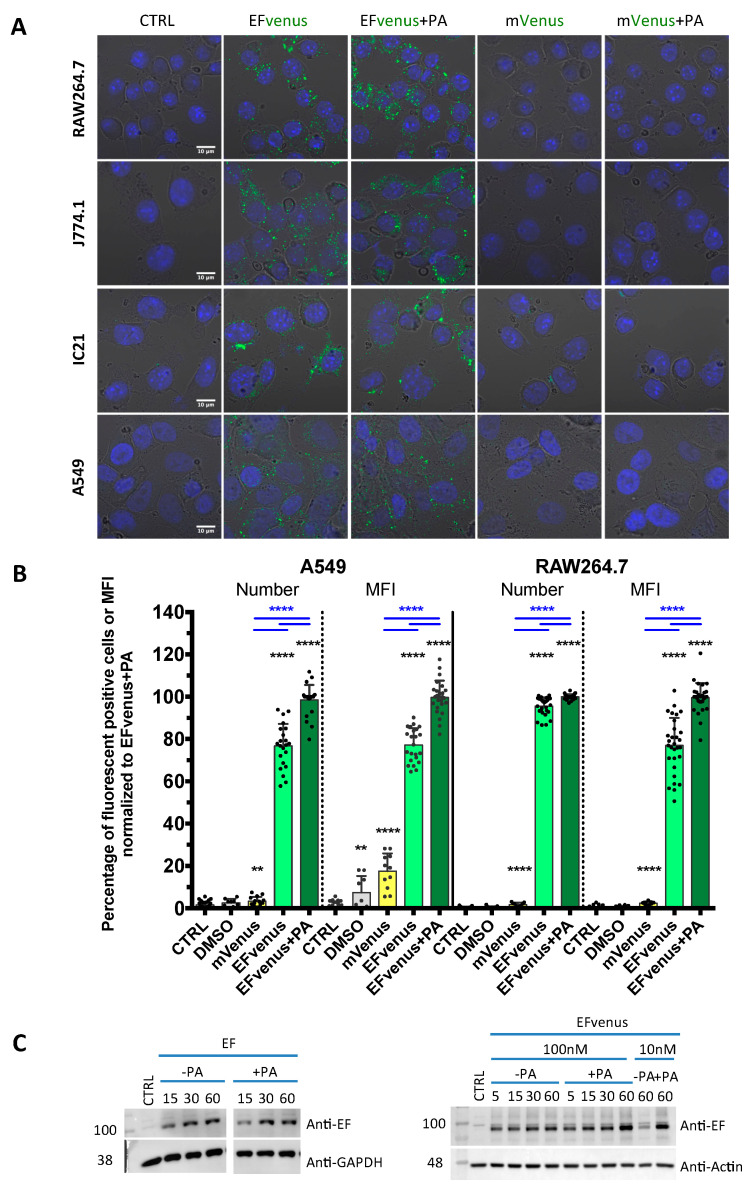
EFvenus is able to enter cells, and the signal is specific. (**A**) RAW264.7, J774.1, and IC21 macrophage cell lines or A549 epithelial cells were exposed for 1 h to 100 nM of mVenus or EFvenus without or with 300 nM of PA at 37 °C. Fluorescence was observed under confocal microscopy with nucleus staining with DAPI (blue), mVenus and EFvenus appearing in green. (**B**) Macrophage cell lines RAW264.7 and epithelial cell line A549 were exposed for 1 h with 100 nM of mVenus or EFvenus, without or with 300 nM of PA. Negative control is cells alone (CTRL) or with 10% DMSO. Fluorescence was observed by flow cytometry for EFvenus (light green), EFvenus + PA (dark green), and mVenus (yellow). Black stars indicate the results of the Mann-Whitney nonparametric test (** *p* < 0.01, **** *p* < 0.0001) comparing each condition to the control. Underlined blue stars indicate a statistical difference in comparison with EFvenus + PA. (**C**) RAW264.7 cells were exposed for 15, 30, or 60 min at 37 °C with 100 nM of EF from List Laboratories or with 10 nM or 100 nM EFvenus, without or with 300 nM of PA. Cell lysates were analyzed by western blot with antibodies against EF (89 kDa), actin (42 kDa), or GAPDH (37 kDa).

**Figure 2 microorganisms-12-00308-f002:**
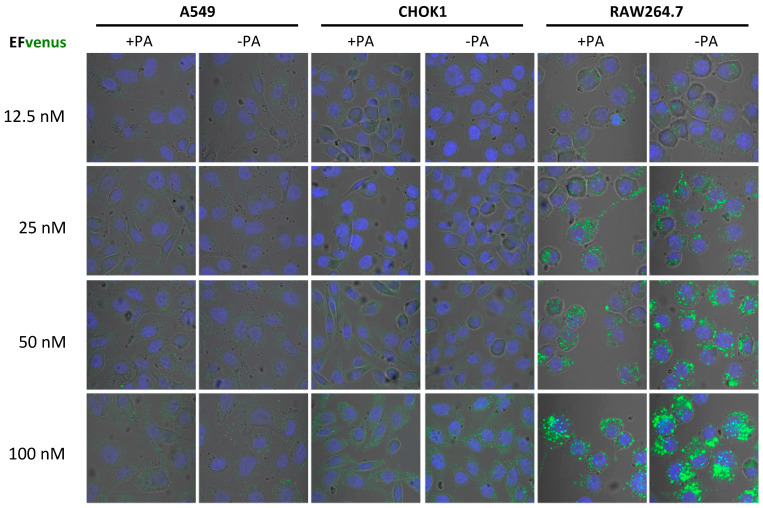
EFvenus entry into cells is detected at various protein concentrations independently of the presence of PA. RAW264.7, CHOK1, and A549 cells were exposed to different concentrations of EFvenus ranging from 12.5 nM to 100 nM, without PA (−PA) or with 300 nM of PA (+PA) for 1 h at 37 °C. Fluorescence was then observed under confocal microscopy. The nucleus appeared in blue, and the EFvenus in green.

**Figure 3 microorganisms-12-00308-f003:**
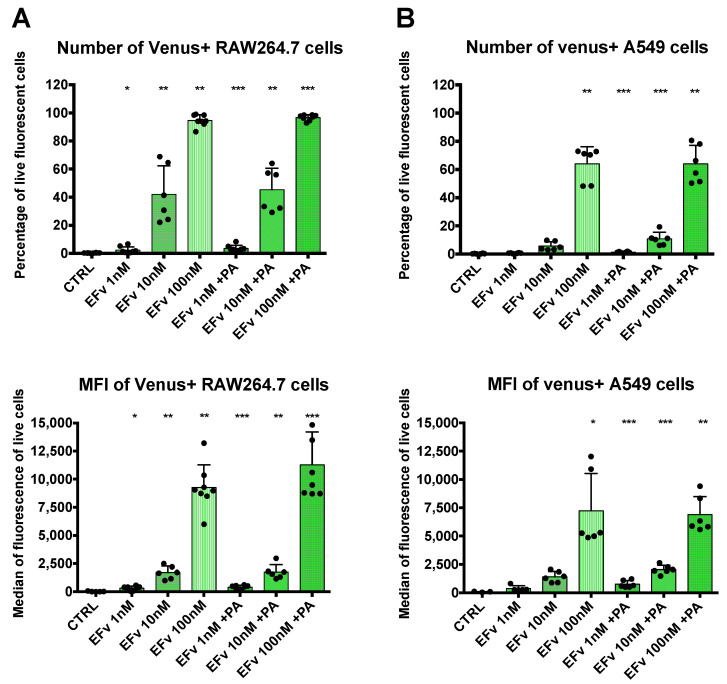
Quantification of EFvenus entry into cells. (**A**) RAW264.7 macrophages or (**B**) A549 epithelial cells were exposed to EFvenus (1 nM, 10 nM, or 100 nM) without or with PA (300 nM) for 1 h at 37 °C. After washing and staining for viability, cells were analyzed by flow cytometry. The number of fluorescent positive cells and the median of fluorescence relative to mVenus are indicated in the top and bottom parts of each graphic, respectively. Data were performed three times in duplicate. Stars represent statistical differences relative to unexposed control cells (CTRL). Statistical analysis was conducted using ANOVA, Kruskall-Wallis multiple comparison test, with * *p* < 0.05, ** *p* < 0.01, *** *p* < 0.001.

**Figure 4 microorganisms-12-00308-f004:**
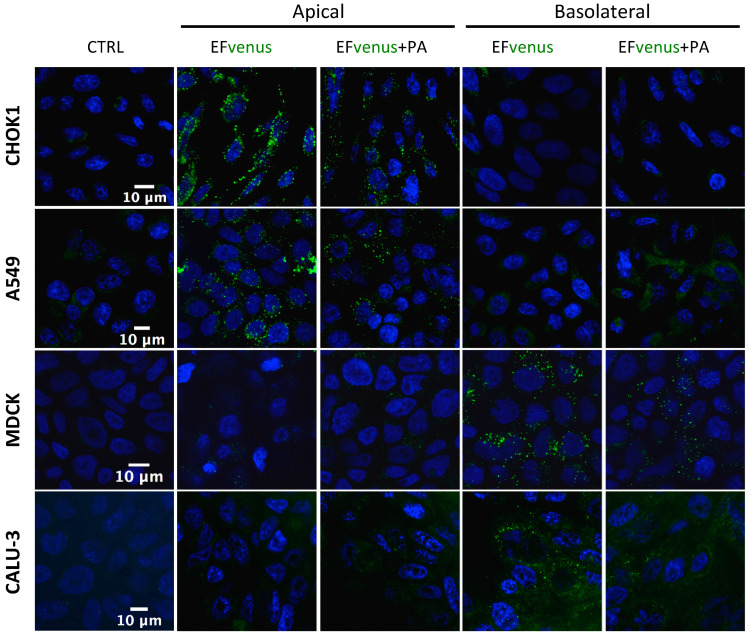
EFvenus entry into epithelial cells is pole dependent. CHOK1, A549, MDCK, and Calu-3 epithelial cell lines were exposed for 1 h at 37 °C to EFvenus (100 nM), with or without PA (300 nM), at the apical or basolateral side. Fluorescence was observed under confocal microscopy, with the nucleus in blue and EFvenus in green, and compared to untreated cells (CTRL for control).

**Figure 5 microorganisms-12-00308-f005:**
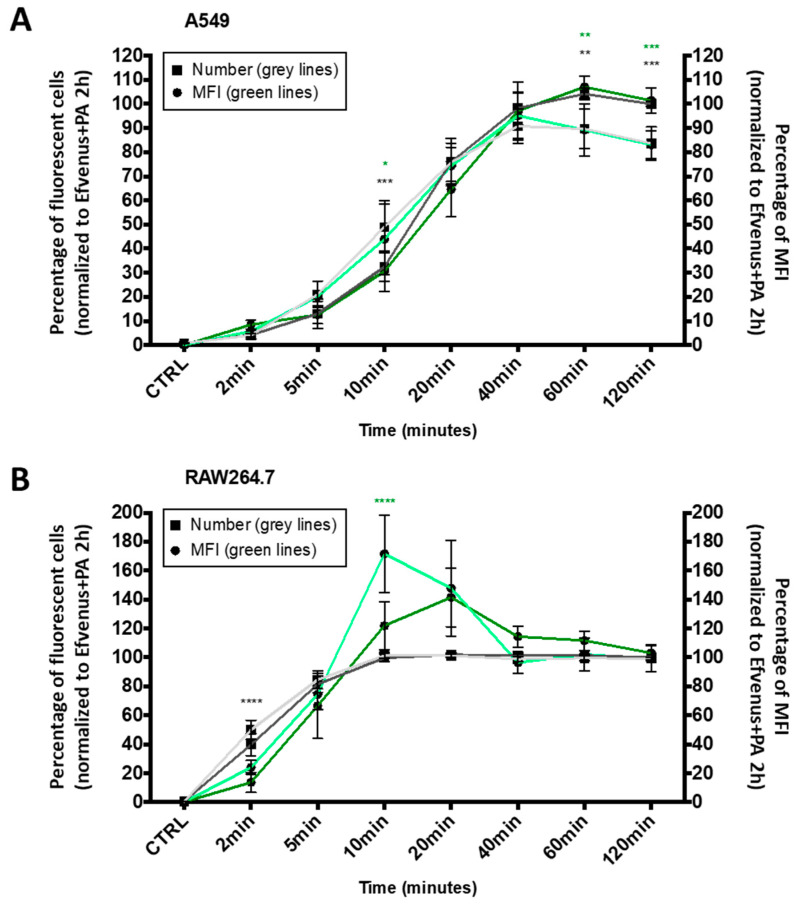
The kinetic profile of EFvenus endocytosis differs between cell types. A549 epithelial cell line (**A**) and RAW264.7 macrophages (**B**) were exposed for 2, 5, 10, 20, 40, 60, or 120 min to 100 nM of EFvenus, with or without PA (300 nM) at 37 °C. The number of fluorescent positive cells, normalized to the condition EFvenus + PA at 120 min as 100%, is indicated with square shapes for EFvenus (light grey) and EFvenus + PA (dark grey). The median of fluorescence (MFI) is indicated with round shapes for EFvenus (light green) and EFvenus + PA (dark green). The percentage of MFI for each time was normalized to the MFI of EFvenus + PA at 120 min, representing 100% of MFI. Data were collected three times in duplicate and were represented as mean ± standard deviation. Stars represent statistical differences between EFvenus and EFvenus + PA for the number of fluorescent cells (black stars) or for the MFI (green stars). Statistical analysis was conducted using two-way ANOVA and Bonferroni multiple comparison test, with * *p* < 0.05, ** *p* < 0.01, *** *p* < 0.001, **** *p* < 0.0001.

**Figure 6 microorganisms-12-00308-f006:**
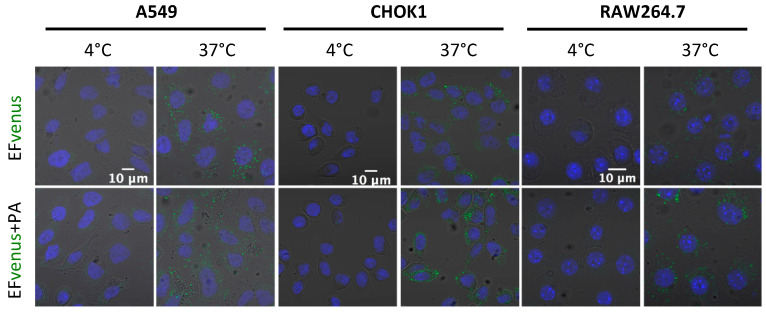
EFvenus entry depends on an active cellular process. A549, CHOK1, and RAW264.7 cells were exposed for 1 h to EFvenus (100 nM), with or without PA (300 nM), at 37 °C (active entry) or 4 °C (passive diffusion). Fluorescence was observed under confocal microscopy, with the nucleus in blue and EFvenus in green.

**Figure 7 microorganisms-12-00308-f007:**
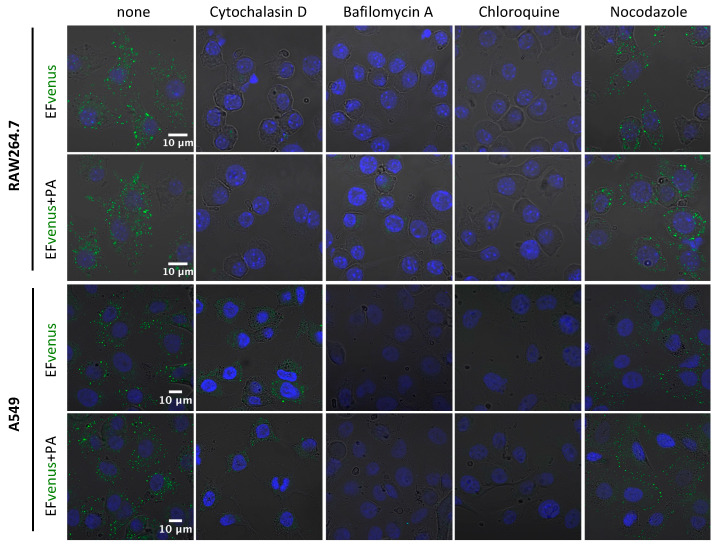
Implication of cytoskeleton and pH of endosomes in EFvenus entry. RAW264.7 or A549 cells were incubated without inhibitor (none) or with cytochalasin D, bafilomycin A, chloroquine, or nocodazole for 30 min at 37 °C and then exposed for 1 h with EFvenus (100 nM) +/− PA (300 nM). Distribution of endocytosed EFvenus was then observed under confocal microscopy with the nucleus in blue and EFvenus in green.

**Figure 8 microorganisms-12-00308-f008:**
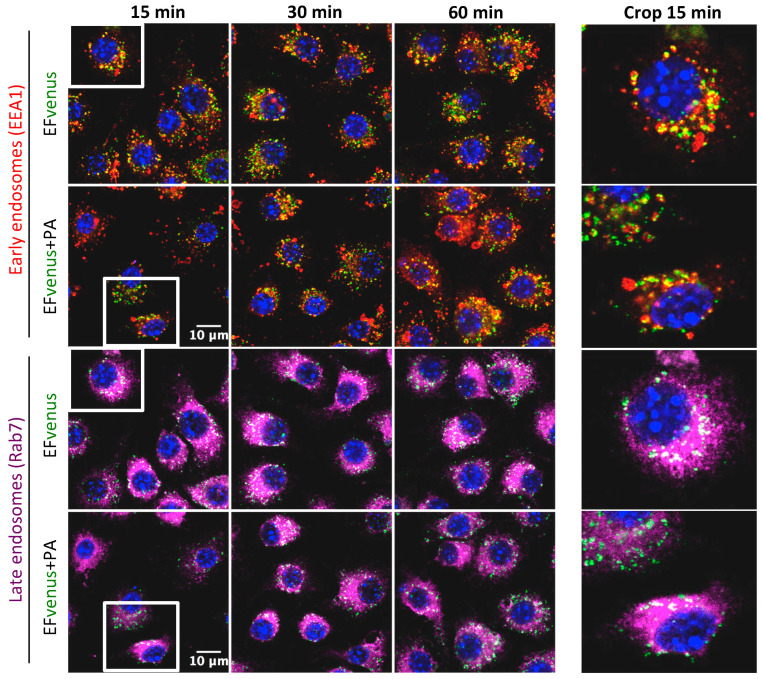
EFvenus follows the same endosomal trafficking as EFvenus + PA in RAW264.7 cells. RAW264.7 macrophages were exposed to EFvenus (green, 100 nM) ± PA (300 nM) for 15, 30, or 60 min at 37 °C. Cells were then stained for the nucleus (blue), early endosomes (anti-EEA1, red), and late endosomes (anti-Rab7, purple). The colocation of EFvenus with EEA1 or Rab7 appeared in yellow and white, respectively. A crop focusing on colocation in one cell at 15 min, corresponding to the white box on the left part of the figure, can be seen on the right part.

**Figure 9 microorganisms-12-00308-f009:**
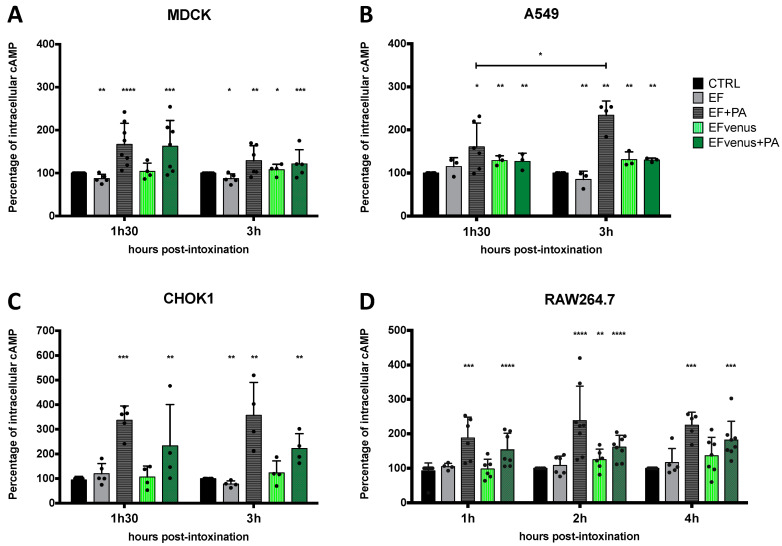
PA is required for the enzymatic activity of EF and fluorescent EF chimera. (**A**) MDCK, (**B**) A549, (**C**) CHOK1, and (**D**) RAW264.7 cells were incubated with 100 nM of EF (gray) or EFvenus (green) ± PA (300 nM) at different times at 37 °C. The level of intracellular cAMP was quantified by ELISA. Results are shown as the mean ± standard deviation (n ≥ 3). Each dot refers to a single experiment. Values were normalized to the control as 100% of the cAMP level. Stars alone indicate the results of the one-way ANOVA nonparametric Dunnett’s multiple comparisons test compared to control. Stars with bars indicate a significant difference between the indicated times for the same condition using the nonparametric Mann-Whitney test, * *p* < 0.05, ** *p* < 0.01, *** *p* < 0.001, **** *p* < 0.0001.

**Figure 10 microorganisms-12-00308-f010:**
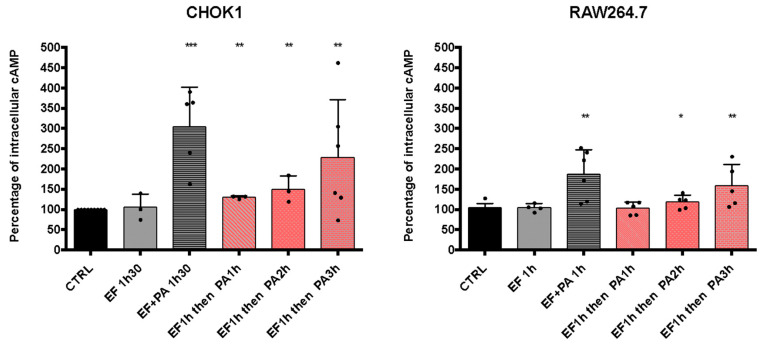
cAMP levels increase after the sequential entry of EF and PA. CHOK1 and RAW264.7 cells were incubated with EF (100 nM) for 1 h at 37 °C and then washed and incubated again for 1, 2, or 3 h with PA (300 nM). The level of intracellular cAMP was monitored by ELISA and compared to that of cells treated with 100 nM of EF ± 300 nM of PA after 90 min of exposure at 37 °C. Results are shown as the mean with SD (n ≥ 3). Each dot refers to a single experiment. Values were normalized to control as 100% of the cAMP level. Stars indicate a significative difference between the control (CTRL, untreated cells) and the indicated condition using the nonparametric Mann-Whitney test; * *p* < 0.05, ** *p* < 0.01, *** *p* < 0.001.

**Table 1 microorganisms-12-00308-t001:** Oligonucleotides used for cloning.

Name	Sequence	Matrix	Target	Resulting Plasmid
Fwd-pQE60	GGATCTCATCACCATCACC	pQE60	pQE60 open with His-tag Cter	pQE60-*cya-mVenus*-His-tag
Rev-pQE60	CATGGTTAATTTCTCCTCTTTAATGAATTCTGTG			
Fwd-(pQE60)-Cya	AAGAGGAGAAATTAACCATGAATGAACATTACACTGAGAG	*B. cereus* G9241 genomic DNA	*Cya* gene	
Rev-Cya-(Fluo+)	TGCTCACCATGAGCTCTTTTTCATCAATAATTTTTTGGAAG			
Fwd-(Cya)-Fluo+	TGATGAAAAAGAGCTCATGGTGAGCAAGGGCGAG	pET28-mVenus	*mVenus*	
Rev-Fluo+-(pQE60)	ATGGTGATGGTGATGAGATCCCTTGTACAGCTCGTCCATGC			
Fwd-pQE60Fluo	GCTGTACAAGTAAGCTTAATTAGCTGAGC	pQE60-EFmVenusHis	pQE60 open with His-Tag Nter	pQE60-His-tag-*cya-mVenus*
Rev-pQE60Cya NterHis	AATGTTCATTGTGATGGTGATGGTGATGCATGGTTAATTTCTCCTCTTTAATG			
Fwd-EFvenus- NterHis	ATTAACCATGCATCACCATCACCATCACAATGAACATTACACTGAGAGTG	pQE60-EFmVenusHis	Cya-mVenus with His-Tag Nter, without His-Tag Cter	
Rev-EFvenus- Nter	ATTAAGCTTACTTGTACAGCTCGTCCATG			

**Table 2 microorganisms-12-00308-t002:** Inhibitors of endocytosis. Their targets and actions have been previously described [63]. Their concentrations for use on RAW264.7 macrophages and A549 cells were determined according to their known action and the percentage of cell viability (Appendix A).

Inhibitors	Targets and Actions	RAW264.7 Macrophages		A549 Epithelial Cells	
		Concentration	Viability	Concentration	Viability
Bafilomycin A	Endosomal acidification (vacuolar H^+^-ATPases)	0.4 µM	92.9%	0.4 µM	98.5%
Chloroquine	Endosomal acidification (maturation)	400 µM	104.3%	500 µM	83.1%
Cytochalasin D	Actin polymerization (most endocytic pathways)	10 µM	93.5%	5 µM	75.7%
Nocodazole	Microtubules depolymerization (trafficking from early to late endosomes)	5 µM	51.2%	10 µM	95.9%

## Data Availability

Data from this study are available upon request.

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
