# Peer review of "Early Circulating Edema Factor in Inhalational Anthrax Infection: Does It Matter?"

_microorganisms, 2024, doi:10.3390/microorganisms12020308_

Round 1

Reviewer 1 Report

Comments and Suggestions for Authors

 The paper has two points: 1) that EF can enter the cell independently of PA and 2) that  EF still requires PA in order to enhance cAMP levels. The first point is relatively important, in terms of how this and similar AB toxin complexes function, while the 2nd point is not new but the authors results suggest it may need some clarification. Previously, the role of PA was considered limited to bringing the toxin into the cell, their results suggest it also directs toxin activation.

The paper is carefully written to illustrate that the authors are really looking at EF entry into the cell on its own with their fluorescent construct. While they show that the mVenus does not give much signal,  the control of using another protein of similar size to EF bound to the tag is not there. Also, there is no real explanation for why PA activates the increase in cAMP.

Confusion arises with the title of Figure 2: it seems to show that yes, there is concentration dependence for EF entry (which is also in the text).

The paper could be improved with a figure at the start illustrating exactly what the authors are cloning. Are they using the whole EF with its prepro region or have they removed the PA binding domain area that is also present in LF? A small figure illustrating this and showing the relative size of the Venus area of EFvenus would help the reader. Also of course showing a control with another test protein with the same fluorescent label behaves differently.  Re the EF assay, direct inhibitors of EF (there are nucleotide and non nucleotide ones) might clarify the role of PA in its activity.

A second point of improvement is the author's wordiness in the overlong discussion, which rambles and brings in points that are not made relevant to the paper at hand. It is as if the authors wanted to combine a review article with their actual results (eg, discussion of sporulation that goes on from 649-707 without any preamble to indicate why it is here). 

Comments on the Quality of English Language

A few typos do not affect the overall readability of the ms but should probably be corrected as well. eg lines 630, 634,717,740 and others. Unprecedented, etc

Reviewer 2 Report

Comments and Suggestions for Authors

Anthrax toxin, produced by Bacillus anthracis and Bacillus cereus is a lethal toxin and an important molecule to understand the intoxification mechanism of the bacterial toxins. In the present work, the authors made a well thought experimental strategies to understand the effect of PA (Protective antigen) on on the entry and activity of EF (Edema Factor). The paper is well written and conclusion is supported by the experimental data. However, minor things should be considered for reevaluation:

1) In table 2, Bafilomycin optimal concentration range?

2) In Figure 1C, spelling of actin?

3) In whole document, concentration is written in 12,5 nM, it should be 12.5 nM?

4) In line 335, it should be "dependent" not "depended"?

5) In line 342, What do you mean by intracellular signal? How it is detected?

6) In figure 4 and its related conclusions, it would be better to quantify the signal with respect to time?

7) In figure 7 and its related conclusions, the experiment should be done with and without cytochalasin D or Nocadazole or both?

8) In line 564, microscopy spelling?

9) Authors should shed a light on possible mechanisms for Anthrax Toxin EF in the absence of PA?

10) In Original WB in figure 1, anti-EF size is 89 kDa or 116 kDa?

Comments on the Quality of English Language

English language is acceptable.

Reviewer 3 Report

Comments and Suggestions for Authors

This is well designed and executed study to evaluate whether Edema factor (EF) can enter target cell lines without the assistance of Protective Antigen (PA), and whether or not it is active in cell intoxication. They have convincingly demonstrated that EF does enter cells independently of PA, although in an inactive form that still requires PA to rescue toxicity, although this can occur after initial EF uptake. The figures are well presented and clear. The discussion focuses on possible mechanisms of EF uptake in the absence of PA, I would suggest some additional discussion of possible mechanisms of EF activation by PA as well (this could be a fruitful pathway for future investigation into small molecule inhibitors). The absence of toxin activity despite uptake in the absence of PA suggests that this is not a primary intoxication mechanism, but rather incidental capture of EF by other non-specific cellular processes, since inactive EF does not produce benefit to the infecting Bacillus anthracis.

Additionally, I have a few specific suggestions listed below.

Line 29  - suggest listing inhalation, cutaneous and gastrointestinal as the 3 clinical forms.

Figure 2 – the legend says “EFvenus entry into cells is not dependent upon toxin concentration”, but the text indicates the opposite - “Moreover, gradual increase in concentration of exposition induced a similar gradual increase in the intracellular signal of EFvenus, regardless of the presence of PA.“ (line 284). The figure also indicates the opposite – there is noticeable EFvenus at the 50nM and 100nM concentrations in CHOK1 cells, but not at 12.5nM or 25nM, and the amount of EFvenus visible in RAW264.7 cells clearly increases with concentration. Possibly this is a typo?

Discussion in general – while EF entry is interesting, given that you demonstrate that in the absence of PA, EF is not enzymatically active and does not produce cAMP, it is unlikely to be important to intoxication. You might want to discuss more about possible hypothesis for how translocation via PA activates EF, as this might be a potential target for small molecule inhibitors that could prevent activation and inhibit toxicity.

Comments on the Quality of English Language

The manuscript does need a moderate amount of editing for English grammatical and spelling errors. 

Ex: Line 28 – “Anthrax is a well-known disease responsible of livestocks epidemics and professional diseases” should be “Anthrax is a well-known disease responsible for livestock epidemics and veterinary diseases” or similar. Numerous minor grammar errors throughout. Note that the word “exposition” should be replaced with “exposure” in multiple uses.
